# DenseFlow: Spotting Cryptocurrency Money Laundering in Ethereum Transaction Graphs

## ABSTRACT

In recent years, money laundering crimes on blockchain, especially on Ethereum, have become increasingly rampant, resulting in substantial losses. The unique features of money laundering on Ethereum, such as decentralization and pseudonymity, pose new challenges for Ethereum anti-money laundering. Specifically, the existence of dense and extensive laundering gangs and intricate multilayered laundering pathways makes it exceptionally challenging for regulators to identify suspicious accounts and trace money flows. To address this issue, we propose an innovative DenseFlow framework that effectively identifies and traces money laundering activities by finding dense subgraphs and applying the maximum flow idea. We conduct multiple experiments on four datasets from Ethereum to validate the effectiveness of our approach. The precision of our DenseFlow is 16.34% higher than the start-of-the-art comparison methods on average, highlighting its distinctive contribution to tackling money laundering issues on blockchain.

## CCS CONCEPTS

• **Applied computing → Electronic funds transfer**; • **Security and privacy → Economics of security and privacy**.

## KEYWORDS

Anti-money laundering, Ethereum, Cryptocurrency, Transaction network, Graph mining

**ACM Reference Format:**
Anonymous Author(s). 2018. DenseFlow: Spotting Cryptocurrency Money Laundering in Ethereum Transaction Graphs. In *Proceedings of The ACM Web Conference (WWW '24)*. ACM, New York, NY, USA, 10 pages. https://doi.org/XXXXXXX.XXXXXXX

## 1 INTRODUCTION

Over the course of the past 15 years, since the concept of Bitcoin [27] being introduced, blockchain technology has attracted sustained global attention and experienced rapid growth. Ethereum, as the pioneering blockchain platform supporting smart contracts [2], has now emerged as the world's second-largest blockchain network, boasting a staggering market capitalization of $19.5 billion. Nevertheless, while blockchain technology presents new opportunities, it is also a breeding ground for a plethora of criminal activities [9, 37], including fund theft, phishing schemes, Ponzi scams, etc. Following

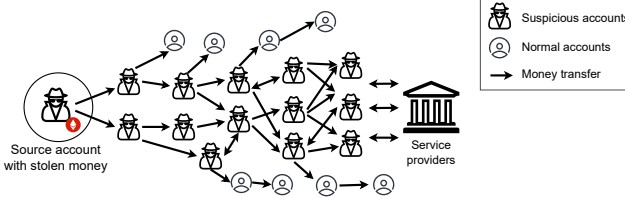

**Figure 1: Example of a cryptocurrency money laundering process on Ethereum. The source account illegally gains money and launders it through dense and complex accounts and transactions until service providers.**

the execution of these crimes, criminals must engage in money laundering to legitimize their ill-gotten gains, allowing them to spend these illegal funds without raising suspicion [31].

Money laundering [30] refers to the act of transforming illegally acquired funds or assets into seemingly legitimate sources through a series of transactions. Currently, illegal activities involving money laundering on blockchain platforms have amassed significant amounts and caused substantial losses, drawing the attention of governments and regulatory authorities. Combating money laundering crimes on blockchain has become an urgent priority. According to reported statistics [31], the amount involved in cryptocurrency money laundering has steadily increased from 2015 to 2022, with a staggering $23.8 billion in losses within a single year in 2022. The cumulative losses have reached an alarming $67 billion. This underscores the severity and rampant nature of money laundering crimes on the blockchain.

Anti-money laundering (AML) in the financial sector is not a new issue, but financial activities on blockchain exhibit significant differences from traditional finance: (1) Decentralized and distributed nature: Blockchain operates on a global network of nodes, free from foreign exchange controls, making it easy to evade financial regulations; (2) Pseudonymity: Users can create unlimited accounts and transactions without the need for real-name verification, making it challenging to trace and track transactions. These characteristics pose significant challenges for regulatory authorities in identifying suspicious accounts and tracing funds:

- **C1: Massive and dense gang of money laundering accounts:** Unlike traditional financial money laundering, where criminals aim to use as few bank accounts as possible [21], in blockchain, creating accounts is nearly cost-free. Criminals can generate numerous disposable accounts, forming massive and dense accounts to launder the proceeds of illegal activities. This strategy helps them evade being frozen by centralized cryptocurrency service providers and escalates the difficulty for regulators to identify suspicious accounts manually.
- **C2: Complex multi-layered money laundering pathways:** To disperse and transfer illegal funds as quickly as possible, hackers create intricate multi-layered money laundering pathways

among numerous accounts. Furthermore, hackers utilize multiple accounts to engage in multiple transactions among themselves, and they may also interact with other types of accounts, such as service provider accounts and regular user accounts, to conceal their activities.

Although there have been some studies on blockchain anomaly analysis and detection [5, 10, 15, 32, 34, 39–41], these methods do not consider finding the core gangs and pathways of money laundering. Most of these paper focuses on account classification, such as phishing account detection [20, 38], Ponzi scheme detection [7], and fraud account classification [14, 17, 26, 35]. This kind of work focuses on the source of the stolen money, and cannot not uncover the money laundering gangs that follow it. Anti-money laundering research in the field of Bitcoin is mainly based on the Elliptic dataset [36] to classify accounts (i.e. illicit accounts vs licit accounts), and this kind of work does not uncover money laundering gangs and money laundering pathways. Anti-money laundering efforts in the Ethereum field are based on heuristic rules to screen suspected money laundering networks [22], but this kind of method is easily circumvented by experienced hackers, resulting in ineffective rules.

To address these challenges, we propose a new framework called **DenseFlow**, designed to assist regulators in identifying and evidencing money laundering activities on the blockchain. To tackle Challenge **C1**, we identify accounts of money laundering gangs engaged in extensive high-frequency asset transfers by detecting **dense** subgraphs and considering the characteristics of money laundering transaction amounts and time sequences. To address Challenge **C2**, we trace the **flow** of funds from money laundering sources to accounts of gangs and supplement involved accounts along the money laundering pathway using maximum flow. We conduct experiments on four real-world incident datasets from the Ethereum platform, comprehensively analyzing the effectiveness of DenseFlow through method comparisons, ablation studies, and case studies.

Overall, our contributions are summarized as follows:

- This paper is the *first* to conduct an in-depth analysis and research on the issue of money laundering detection on Ethereum and introduce the DenseFlow method innovatively.
- We design the suspiciousness metric for accounts and transactions based on the traits of money laundering behavior, rather than relying on a black-box model. Furthermore, we prove that our algorithm approaches the theoretical boundaries of approximate optimality in detecting money laundering within a graph.
- We implement the DenseFlow method[1] and demonstrate its superior effectiveness via multiple experiments. The precision of our DenseFlow is 16.34% higher than the start-of-art comparison methods on average.

The remainder of this paper is organized as follows. Section 2 gives the background related to our approach, and Section 3 surveys the related work. Section 4 defines and formulates the Ethereum money laundering detection problem. Section 5 introduces the framework of DenseFlow. Section 6 presents an extensive experimental evaluation to validate the effectiveness of our framework.

---

[1]Available at https://github.com/DenseFlow.

Section 7 finally concludes this work. Additional theoretical proofs are provided in the appendix.

## 2 BACKGROUND

### 2.1 Money Laundering

Money laundering involves disguising financial assets so they can be used without detecting the illegal activity that produced them, through money laundering, the criminal transforms the monetary proceeds derived from criminal activity into funds with an apparently legal source [8]. Money laundering facilitates a broad range of underlying severe criminal offenses and ultimately threatens the integrity of the financial system [33].

Traditionally, money launderers engaging in layering repeatedly move fiat currency into different financial institutions and assets to blur the origins of the criminal proceeds. Figure 1 demonstrates an example of the Ethereum money laundering process. With crypto, money launderers may move the illicit funds through hundreds of wallets before depositing the funds and cashing out the funds at service providers[2], e.g. crypto exchange. Unlike bank accounts, thousands of wallets may be opened without proof of identity, within seconds [18].

### 2.2 Ethereum and Transactions

Ethereum is the first blockchain platform that supports smart contracts. On Ethereum, users can create and execute smart contracts, which are computer programs designed to enforce the terms of a contract automatically. Ethereum transactions involve financial activities conducted on the Ethereum blockchain, similar to transactions in traditional financial systems. These transactions encompass the transfer of the digital currency Ether, and may involve actions such as transferring funds from one account to another, executing smart contracts, or engaging in other operations related to digital assets [6]. Each transaction is recorded on the Ethereum blockchain, ensuring transparency and immutability of the transaction history. Ethereum's transaction system offers the financial sector decentralized and secure alternatives, enhancing the efficiency and traceability of various financial activities.

## 3 RELATED WORK

### 3.1 Anti Money Laundering

In the traditional financial anti-money laundering research, Li *et al.* [21] proposed a graph-based method to detect money laundering behavior. Sun *et al.* [28] proposed a model to detect money laundering agent accounts by keeping track of their residuals and other features. In the study of blockchain financial anti-money laundering, Alarab *et al.* [1] used an ensemble learning approach to detect money laundering transactions on the Bitcoin blockchain. Lorenz *et al.* [25] proposed active learning and unsupervised methods to detect money laundering activities on a Bitcoin transaction dataset. Lin *et al.* [22] mined untagged money laundering gangs on Ethereum through heuristic transaction tracking methods, to carve out a complete picture of security incidents.

---

[2]Typically, the conversion of virtual currency to fiat currency by hackers requires navigating through one or a series of centralized virtual asset service providers like exchanges or over the counter (OTC). We use the term "service providers" to denote a group of virtual currency service providers that adhere to AML compliance.

## 3.2 Graph-based Anomaly Detection

Graph-based techniques excel at identifying groups of fraudsters, typically by identifying irregularities through density indicators. Evading such detection is challenging because fraudulent activities inherently involve interconnected relationships, represented as edges within the graph. In recent years, many studies have used graph-based methods to accomplish anomaly detection and demonstrated their effectiveness. HoloScope [24] introduced contrast suspiciousness from graph topology and spikes to accurately detect fraudulent users and objects. FRAUDAR [13] proposed a camouflage-resistant weighting scheme to calculate the amount of fraud adversaries can have, even in the face of camouflage. ANTICO [16] presented a class of metrics to capture suspicious signals of the activities and a greedy algorithm to spot suspicious dense subgraphs by optimizing the proposed metric. AntiBenford [4] proposed a novel unsupervised framework for detecting anomalies in financial networks.

## 4 PROBLEM FORMULATION

On Ethereum, after successful fund theft, hackers strategically create a multitude of intermediary accounts to obscure the flow of funds. They engage in multi-layered and complex transactions, facilitating the transfer of stolen funds from the source account to several accounts.

### 4.1 Analysis

We assume the following measurable traits in money laundering activities on Ethereum.

Trait 1 (dense transfer). *Hackers construct a large and dense subgraph of money transfers within gang accounts.*

In the realm of blockchain, all transactions are publicly transparent. Therefore, large-scale asset theft will attract significant attention from monitoring systems. To circumvent potential freezing by service providers, hackers urgently engage in frequent and substantial fund transfers within gang accounts but have limited collaboration with other accounts. This results in the density of the money laundering transfer subgraph, particularly during the latter stages of money laundering, i.e., the process of converting assets into fiat currency.

Trait 2 (Temporal Surge). *Hackers tend to engage in frequent fund transfers within a short timeframe.*

This trend arises from hackers conducting multiple rounds of illicit fund transfers, resulting in a surge in transaction activity within money laundering accounts. Conversely, hackers typically opt for one-time use of these accounts to evade detection by service providers. This leads to the termination of the money laundering account's lifecycle corresponding to a sudden decline in its transaction activity. Therefore, the behavior of money laundering accounts often manifests as peaks in the temporal sequence.

Trait 3 (Amount Intensity). *The proportion of suspicious transactions in a specific time period significantly exceeds that of other time periods.*

Due to the urgent need to transfer substantial illicit assets in the hands of hackers, frequent transactions occur during the money laundering period, resulting in an uneven temporal pattern of transaction amounts. Leveraging this trait, we define a suspiciousness rating for each transaction (refer to Section 5 for details).

Trait 4 (Rating Deviation). *Money laundering accounts show a significant difference in the suspiciousness score of transactions compared with regular accounts.*

This is attributed to the fact that money laundering accounts are specifically employed by hackers for the purpose of money laundering, rendering the majority of their transactions suspicious. In contrast, transactions by regular users exhibit greater randomness, leading to a noticeable deviation in the suspiciousness rating compared to money laundering accounts.

In addition, Trait 1 is related to a dense topological structure. Trait 2 is associated with temporal behavior. Trait 3 is correlated with both temporal behavior and transaction amount. Trait 4 is linked to rating deviation. In summary, hackers engage in money laundering activities by rapidly and intensively transferring embezzled funds through multiple layers from the source account. This results in a substantial cash flow, which distinguishes from ordinary transactions.

### 4.2 Definitions and Notations

Generally, the source account responsible for the stolen assets in each incident can be obtained from incident reports. We can acquire a biased downstream multi-layered transaction history starting from the incident's source account by Lin's tool [22]. Thus, we use this biased downstream transaction record of the hacker as input. The transaction records $(i, j, a, t)$ include the sender's account $(i)$, the recipient's account $(j)$, the transaction amount in Ether $(a)$, and the timestamp $(t)$ indicating the time when the transaction occurred.

To explore the relationship of transactions between accounts, we model transaction records as graphs [19, 23, 42]. We model the input hacker's downstream transactions as a directed graph $G = (V, E)$, where $V$ represents accounts (including both money laundering (ML) and non-money laundering accounts). $E$ represents directed transactions (multiple transactions may occur between each pair of accounts), and $e_{ij}$ represents the frequency from node $i$ to node $j$.

Problem (Ethereum ML Detection): *Given a downstream transaction network $G$ starting from the source account of an Ethereum incident,*

- *Identify a dense subgraph in $G$ consisting of suspicious money laundering gang accounts;*
- *Optimize suspiciousness metrics under general cryptocurrency money laundering knowledge, considering topology, transaction time, and amount;*
- *Trace the flow of funds from the source to money laundering accounts and supplement on group accounts.*

Table 1 displays the symbols primarily used throughout the entire paper.

## 5 METHODOLOGY

We design DenseFlow for Ethereum money laundering detection, targeting suspicious account gangs and money laundering flows. Firstly, we collect the downstream transaction network $G$ starting

**Table 1: Symbols and definitions**

| Symbols | Interpretation |
|---|---|
| $G$ | Downstream transaction network |
| $V$ | Nodes of graph $G$ |
| $E$ | Transaction edges of graph $G$, each edge $e = (i, j, a, t)$ |
| $a$ | Transaction amount |
| $t$ | Transaction timestamp |
| $e_{ji}$ | Edge frequency from node $j$ to node $i$ |
| $S$ | Node subset of graph $G$ |
| $\alpha_i(S)$ | Topological suspiciousness of node $i$ within $S$ |
| $\beta_i(S)$ | Temporal suspiciousness of node $i$ within $S$ |
| $\gamma_i(S)$ | Monetary suspiciousness of node $i$ within $S$ |
| $\Phi[T_i(S)]$ | Timestamp set of transaction from $S$ to $v_i$ |
| $R$ | Rating of transaction suspiciousness |
| $\omega_i$ | Weighted assigned to node $i$ in priority tree |
| $F$ | Accounts within suspicious money laundering flow |
| $M$ | Suspicious money laundering account set |
| $f_i(S)$ | Suspicious function of node $i$ within $S$ |
| $g(S)$ | Suspicious function of subset $S$ |

from the source hacker accounts, and then model it as a transaction graph. Subsequently, within $G$, we identify suspicious account gangs through proposed suspiciousness metrics and dense subgraph algorithm. These gangs exhibit the key traits of generating high-frequency and dense transactions to launder illicit gains. Then, based on the account of gangs and the source account, we employ the maximum flow algorithm to trace the money laundering pathway between the source account and the suspicious gangs. In the end, DenseFlow generates the money laundering subgraph for the Ethereum incidents, including the gangs of accounts and laundering flows.

## 5.1 Proposed Suspiciousness Metric

Given this problem definition, we propose how to measure the ML suspiciousness of subgraphs in $G$. According to the key traits discussed in Problem Formulation, we propose three metrics for optimization.

*Topological suspiciousness.* According to Trait 1, hackers construct a large and dense subgraph of money transfers within gang accounts but have limited transactions with other accounts. Thus, an account becomes more suspicious when it primarily establishes transactions with suspicious accounts and has fewer connections with others. This can be expressed mathematically by defining the topological suspiciousness of node $i$ within a suspicious subset $S$

$$\alpha_i(S) = \frac{\sum_{(j,i) \in E \wedge j \in S} e_{ji}}{\sum_{(k,i) \in E \wedge k \in N} e_{ki}}, \tag{1}$$

where $e_{ji}$ denotes the transaction frequency from node $j$ to node $i$. $\alpha_i(S)$ measures the density and involvement ratio of node $i$ within subset $S$.

By dynamically adjusting the topological suspiciousness of node $i$, DenseFlow can improve the accuracy of ML detection in "noisy" graphs, even with low ML density.

*Temporal suspiciousness.* According to Trait 2, the behavior of money laundering accounts often manifests as peaks in the temporal sequence. Considering the burst and drop patterns described in Trait 2, we first need to detect surge awakening and peak points of the temporal sequence. By the MultiBurst algorithm[24], The awakening point and peak points are calculated and represented as $(t_o, a_o)$ and $(t_p, a_p)$, respectively. $t$ denotes the transaction timestamp and $a$ denotes the transaction amount.

The temporal suspiciousness ($\beta$) of a node $i$ is represented by the ratio of the node's participation level ($\Phi$) in certain transaction surges within the subset $S$ compared to the entire node set $V$. The temporal suspiciousness of a node $i$ is designed as

$$\beta_i(S) = \frac{\Phi[T_i(S)]}{\Phi[T_i(V)]}, \tag{2}$$

where $T_S^i$ denotes the transaction timestamp set of node $i$ connecting within nodes of the subset $S$. The participation level ($\Phi$) of node $i$ within subset $S$ can be quantified by dividing the intensity of the surge by the frequency with which it occurs, and the intensity calculation entails multiplying the slope by the surge's increment.

$$\Phi[T_i(S)] = \sum_{(t_o, t_p)} k_{op} \cdot \Delta a_{op} \sum_{t \in T_S^i} \mathbb{I}(t \in [t_o, t_p]), \tag{3}$$

where $\Delta a_{op} = a_p - a_o$, $k_{op}$ denotes the slope of the line going through $(t_o, a_o)$ and $(t_p, a_p)$, and $\mathbb{I}$ is the indicating function.

With the temporal suspiciousness, DenseFlow can measure the temporal surge pattern of money laundering behavior. Noticeably, the method has the capability to adjust bin sizes tailored to different nodes. For instance, accounts with large transactions benefit from finer bins, facilitating the exploration of intricate patterns.

*Monetary suspiciousness.* According to Trait 3, if there is a sudden increase in the transfer amount of a transaction within a certain period, we consider the transaction as suspicious. We can consider the suspicious rating of a transaction to be related to the ratio of its total transaction amount within a specific time period to the total amount over the entire duration. We slice the timestamp set of the transaction graph $G$ based on a month-long duration. For each transaction, we calculate the total amount of all transactions within each slice. $A(\tau)$ denotes the sum of the amounts for all transactions within the timestamp set $\tau$. $\tau(t)$ represents the timestamp set of this transaction timestamp $t$ within the time slot.

The calculation of the suspicious rating $R(i, j, a, t)$ of a transaction $(i, j, a, t)$ can be performed by

$$R(i, j, a, t) = \frac{A([t - d, t + d])}{A(\tau(t))}, \tag{4}$$

where $[t - d, t + d]$ refers to the collection of transaction timestamps within the two days before and after the transaction timestamp $t$.

According to Trait 4, we know that money laundering accounts have many suspicious transactions compared to regular accounts, showing differences in the rating distribution. In order to quantify the difference in transaction rating distributions, we employ KL-divergence ($KL$) to calculate the rating distribution of node $i$ in the subset $S$ compared to the rest of the nodes (i.e., the complement $V \setminus S$). The monetary suspiciousness score ($\gamma$) of node $i$ in subset $S$

is obtained by weighting the KL-divergence with a balancing factor

$$bal = min\{\frac{\eta_i(S)}{\eta_i(V\backslash S)}, \frac{\eta_i(V\backslash S)}{\eta_i(S)}\}, \text{and } \eta_i(S) = \sum_{(j,i)\in E \wedge j\in S} e_{ji}, \quad (5)$$

$$\gamma_i(S) = bal * KL\left[R_i(S)), R_i(V\backslash S)\right], \quad (6)$$

where $R_i(S)$ refers to the transaction rating set of node $i$ within subset $S$. We use the complement $V\backslash S$ to measure distribution differences, rather than the entire set $V$, to avoid trivial cases where the majority of transaction ratings come from $S$.

*Suspiciousness fusion.* In order to comprehensively leverage different metrics, including topological, temporal, and monetary suspiciousness, we need a method to aggregate these metrics. The most effective approach is the natural joint probability method, which involves multiplying these metrics together:

$$f_i(S) = \sum_{(j,i)\in E \wedge j\in S} e_{ji} \cdot b^{\alpha_i(S)+\beta_i(S)+\gamma_i(S)-3}, \quad (7)$$

where $f_i(S)$ refers to the joint suspiciousness of a node $i$ to subset $S$, $b$ refers to a hyperparameter. In such a way, DenseFlow can dynamically update the node's suspiciousness while the suspicious subset $S$ is evolving. The total suspiciousness ($g$) of subset $S$ of graph $G$ can be defined based on the node suspiciousness

$$g(S) = \frac{\sum_{i\in S} f_i(S)}{|S|}. \quad (8)$$

## 5.2 Greedy Approximation Algorithm

Given the problem definition and suspiciousness of the subset, we need to find subset $S$ that maximizes the objective $g(S)$ in Eq. (8).

Inspired by Charikar's greedy peeling method [3], DenseFlow iteratively identifies and removes the node with the minimum suspiciousness and updates the suspiciousness of related nodes. Each iteration involves traversing all nodes to locate the node with the minimum suspiciousness, incurring significant computational overhead. Therefore, we first construct a priority tree for the nodes in subset $S$.

The leaf nodes of the priority tree represent nodes in $S$, and each internal tree node records the minimum value among its child nodes. The minimum priority tree is designed to efficiently locate the leaf node corresponding to the global minimum value recorded at the root node. This approach reduces the time complexity from $O(|S|)$ to $O(log|S|)$. The weight (i.e., priority) of subset $S$ assigned to node $i$ is defined as

$$\omega_i(S) = f_i(S). \quad (9)$$

We employ an approximate greedy algorithm based on priority trees as shown in Algorithm 1. DenseFlow takes the directed graph $G$ as input and initializes subset $S$ starts at the whole node set $N$. In each iteration, we calculate the suspiciousness for each node based on metrics (1) (2), (6) and weight of priority tree (9). Then, we remove the node in $S$ with minimum weight, approximately maximizing objective (8), and update the weight of its connected nodes. Let $S^{(x)}$ denote the nodes of the subgraph at the $x$-th iteration, and iteratively generate a decreasing sequence of node subsets $S^{(0)} \supset S^{(1)} \supset ...$, and so on until the remaining part is empty.

Finally, the DenseFlow algorithm outputs the final subset $S^*$ with the maximum suspiciousness $g(S)$.

---

**Algorithm 1** Pseudocode of Dense Subgraph Detection

---

**Require:** $G = (V, E)$ a directed graph
**Ensure:** Optimal subset $S^*$
 1: $S \leftarrow N$
 2: $\omega_i \leftarrow$ Calculate node weight as Eq. (9)
 3: $P \leftarrow N$ Build priority tree for $S$ with $\omega_i(S)$
 4: **while** $S$ is not empty **do**
 5:     $i \leftarrow$ Find the minimum weighted node in $P$
 6:     $S \leftarrow S\backslash\{i\}$
 7:     Update priority tree $P$ for all neighbors of $i$
 8:     $g(S) \leftarrow$ calculate as Eq. (8)
 9: **end while**
10: Return subset $S^*$ that maximizes $g$ during the loop.

---

## 5.3 Tracing Maximum Flow

After the suspicious subset detection, we have obtained multiple subgraphs representing money laundering gangs. Naturally, we aim to identify the money laundering pathway connecting the source account to these money laundering gangs. In a directed transaction graph, nodes represent accounts, and the weights on the edges indicate transaction amounts. Given a source account and downstream money laundering gangs, can we find the flow of money laundering funds?

We consider using the maximum flow algorithm [11] to discover the money laundering pathways. The maximum flow problem is to find the maximum possible flow from the source node to the junction in a directed graph. This problem is often used to simulate the flow of fluids in a network or other applications in computer science and operations research. The mathematical form of a simple maximum flow problem can be expressed as follows:

$$\max \sum_{(i,j)\in E} x_{ij}, \quad (10)$$

where $x_{ij}$ is the flow on edge $(i, j)$. Subject to the following constraints:

(1) capacity constraint. $0 < x_{ij} < Capacity_{ij}$, and $Capacity$ refers to the transaction amounts in our problem;

(2) flow conservation constraint. For all nodes $k$ (except the source and sink)

$$\sum_{(i,k)\in E} x_{ik} = \sum_{(k,j)\in E} x_{kj};$$

(3) source and sink flow constraints. For the source node $s$,

$$\sum_{(s,j)\in E} x_{sj} - \sum_{(j,s)\in E} x_{js} = Supply_s,$$

and for the sink node $t$,

$$\sum_{(i,t)\in E} x_{it} - \sum_{(t,i)} = Supply_t$$

where $Supply_s$ and $Supply_t$ are the total inflow or outflow of the source and sink nodes, respectively. Google OR-Tool [12] is used to solve the optimization problem.

This part traces the flow direction of funds involved in money laundering from the source node to various money laundering

nodes. Let account set $F$ be the accounts within the identified maximine flow. Our final suspicious money laundering account set is defined as $M = S^* \cup F$.

## 5.4 Theoretical Analysis

In this part, we discuss the bounds of the optimal solution for our algorithm, providing a theoretical lower bound for our algorithm and the associated proofs.

To make a clear explanation, we define $f(S)$ as the overall node suspiciousness of subset $S$:

$$f(S) = \sum_{i \in S} f_i(S)$$

LEMMA 1. *Consider a subset $S$, and remove node $i_0$ from $S$, let $S' = S/\{i_0\}$, then we can draw a conclusion that:*

$$f(S') \geq f(S) - f_{i_0}(S) \tag{11}$$

PROOF. It is obvious that, if $S' \in S$ then $f_i(S) \geq f_i(S')$ because

$$f(S) = \sum_{i \in S} f_i(S) = \sum_{i \in S'} f_i(S) + f_{i_0}(S)$$
$$\geq \sum_{i \in S'} f_i(S') + f_{i_0}(S')$$

□

LEMMA 2. *For optimal subset $S^*$ and any node $i \in S^*$:*

$$f_i(S^*) \geq g(S^*).$$

The proof of Lemma2 is presented in the Appendix.

THEOREM 1 (BOUND OF OPTIMAL SOLUTION). *Set $g(S^*)$ be the theoretical optimal solution of greedy algorithm, and suppose $g(S^{(k)})$ be the output of algorithm. Then*

$$g(S^{(k)}) \geq \frac{b-1}{b^2} g(S^*),$$

*where $b$ represents the base of $f_i(S)$ in Eq. (7).*

PROOF. Without loss of generality, let $S^* = S^{(k)}$, and denote $S^{(k-1)}$ as the last subset before $S^{(k)}$. $i_1$ represents the node that is removed at the last step, that is $S^* = S^{(k-1)} \backslash \{i_1\}$. For each node $i \in S^*$:

$$f_i(S^{(k-1)}) > f_i(S^*), f_{i_1}(S^{(k-1)}) > f_i(S^{(k-1)})$$
$$g(S^{(k-1)}) \geq \frac{(b-1)}{b^2} f_{i_1}(S^{(k-1)}) \tag{12}$$

**Table 2: Statistics of evaluation datasets. '#' means the number. 'Heists' means the number of accounts labeled as suspicious money laundering accounts in the dataset. The time span is shown in days.**

| Dataset | # Accounts | # Transactions | # Heists | Time |
|---|---|---|---|---|
| PlusTokenPonzi | 34,521 | 58,049 | 30,782 | 817 |
| AlphahomaraExploit | 76,130 | 612,349 | 6,960 | 2494 |
| CryptopiaHack | 152,779 | 815,242 | 8,787 | 2,481 |
| UpbitHack | 377,912 | 1,627,861 | 16,533 | 2,310 |

The proof of inequality (12) will be presented in the Appendix due to lack of space.

$$g(S^{(k)}) \geq g(S^{(k-1)}) \geq \frac{(b-1)}{b^2} f_{i_1}(S^{(k-1)})$$
$$\geq \frac{(b-1)}{b^2} f_{i_1}(S^*) \geq \frac{b-1}{b^2} g(S^*)$$

□

## 6 EVALUATION

In this section, we perform experiments to demonstrate the effectiveness of the proposed DenseFlow. In particular, we aim to answer the following research questions (RQ):

- **RQ1: Model effectiveness.** Can our method outperform existing dense subgraph detection algorithms?
- **RQ2: Proposed metric assessment.** For different datasets, on which metrics do algorithms perform better?
- **RQ3: Case study.** Can our method effectively trace the money laundering behavior in practical use?

## 6.1 Settings

*6.1.1 Dataset.* We choose four money laundering datasets PlusTokenPonzi[3], AlphahomoraExploit[4], CryptopiaHack[5], UpbitHack[6], which are real-world incidents and collected based on the work of Lin *et al.* [22], using a heuristic rule-based approach. Table 2 summarizes the essential characteristics of the four money laundering datasets employed in this study, encompassing the total number of accounts, transactions, transaction amounts, the count of those marked as "Heist" accounts, and the temporal span of the dataset, including normal and suspicious transactions.

It is worth noting that the labels of the heuristic rule-based approach are obtained from the taint analysis and the taint propagation depends on the setting of the hyperparameters. The parameters are set loosely so there may be redundancy in accounts labeled as "Heist" accounts, meaning not all accounts marked as "Heist" are necessarily involved in money laundering. Our goal is to cover as many implicated funds as possible with as few nodes as necessary, thereby enhancing the efficiency of regulatory scrutiny.

*6.1.2 Comparison methods.* We compare our proposed DenseFlow framework with start-of-art graph-based anomaly detection via dense subgraph.

- **FRAUDAR [13].** FRAUDAR detects fraud attacks in graphs by a novel family of suspiciousness metrics that satisfy intuitive traits.
- **Cubeflow [29].** Cubeflow is a flow-based approach to spot fraud from a mass of transactions and proposes a multi-attribute metric for money-laundering flow.
- **Holoscope [24].** Holoscope introduces the "contrast suspiciousness" metric, aiming to detect fraudulent users and entities by integrating the topological structure and peak information on the graph.

[3]https://etherscan.io/address/0xf4a2eff88a408ff4c4550148151c33c93442619e
[4]https://etherscan.io/address/0x905315602ed9a854e325f692ff82f58799beab57
[5]https://etherscan.io/address/0xc8b759860149542a98a3eb57c14aadf59d6d89b9
[6]https://etherscan.io/address/0xa09871aeadf4994ca12f5c0b6056bbd1d343c029

**Table 3: Performance comparison results on four datasets. '-' means never returning a possible result due to the size of datasets. 'DenseFlow' means our method considering all the suspiciousness metrics, and 'DenseFlow*' means our methods with the best result. MCR means money cover rate, and |M| means the number of identified laundering accounts.**

| Dataset | PlusTokenPonzi | | | AlphahomaraExploit | | | CryptopiaHack | | | UpbitHack | | |
|---|---|---|---|---|---|---|---|---|---|---|---|---|
| Metric | Precision | MCR | \|M\| | Precision | MCR | \|M\| | Precision | MCR | \|M\| | Precision | MCR | \|M\| |
| FRAUDAR | 0.9167 | 0.03 | 60 | 0.5122 | 0.32 | 82 | 0.5217 | 0.0004 | 46 | 0.5083 | 0.11 | 120 |
| Cubeflow | 0.2841 | 0.19 | 5,174 | 0.0300 | 0.31 | 15,951 | - | - | - | - | - | - |
| HoloScope | 0.2423 | 0.11 | 549 | 0.1786 | 0.0001 | 17 | 0.1529 | 0.0001 | 14 | 0.00 | 0.01 | 15 |
| DenseFlow | 0.9912 | **0.83** | 19,402 | 0.4929 | 0.33 | 211 | 0.5282 | **1.00** | 426 | 0.6856 | 0.39 | 264 |
| DenseFlow* | **0.9981** | 0.78 | 16,641 | **0.5579** | **0.52** | 656 | **0.6049** | **1.00** | 329 | **0.6994** | **0.50** | 326 |

*6.1.3 Evaluation metrics.* In this paper, we employ the following three validation metrics to comprehensively evaluate the performance of different methods for Ethereum money laundering detection:

- Precision. The precision rate means the percentage of real money laundering nodes in the identified suspicious accounts.

$$Precision = \frac{TP}{TP + FP},$$

where $TP$ is true positives and $FP$ is false positives.

- Money coverage ratio (MCR). MCR refers to the proportion of money involved in the subset of identified money laundering accounts compared to the total amount associated with labeled money laundering accounts.

$$MCR = \frac{\sum_{(i,j)\in E \wedge i\in M_{pred}} a_{ij}}{\sum_{(i,k)\in E \wedge k\in M_{true}} a_{ik}},$$

where $M_{true}$ denotes the set of accounts labeled as "Heist", and $M_{pred}$ is the predicted suspicious account by the algorithms.

- The node number of suspicious account set (|M|). We consider this metric to investigate whether our proposed method can identify core team accounts.

It is worth mentioning again that since the accounts labeled as "Hesit" in the dataset may not be suspicious, therefore the Recall metric is not considered in this paper in favor of the Precision metric.

## 6.2 RQ1: Model Effectiveness

To address RQ1, we evaluate the performance of compared methods in Ethereum money laundering detection. The corresponding results are shown in Table 3. The following conclusions can be drawn:

(1) Precision considers how many of the money laundering accounts detected by the algorithm are indeed labeled as money laundering. Our precision exceeds the optimal results of the compared methods on the four datasets by 8.14%, 29.77%, 8.32%, and 19.11%, respectively. It shows that our algorithm can perform better in money laundering detection accuracy. In particular, Cubeflow does not run out of results for both the CryptopiaHack and UpbitHack datasets, which have a large number of accounts and transactions.

(2) MCR (Money Coverage Ratio) assesses the extent to which the algorithm's detection of money laundering accounts covers the labeled money laundering funds in the dataset. This evaluates whether the algorithm accurately detects core money laundering accounts, which typically involve more dense and frequent money laundering activities and are the primary focus of detection and evidence collection for regulatory authorities. Our MCR significantly outperforms existing methods across four datasets, especially achieving 83% and 100% on the PlusToken and CryptopiaHack datasets, respectively.

(3) The number of suspicious nodes (|M|) indicates how well the algorithm can detect as few nodes as possible while achieving high precision and coverage. This is because a large number of suspicious annotations can inconvenience regulatory personnel in subsequent manual reviews and evidence organization. Compared to the "Heist" accounts labeled in the dataset, each algorithm has significantly reduced the number of suspicious nodes. However, the existing algorithms may have reduced the quantity excessively, which could be one of the reasons for the low MCR.

We further analyze the reasons behind the superiority of our method over the comparison methods. The FRAUDAR method, based on an unweighted graph, primarily relies on the density of node degrees, resulting in the inability to accurately identify money laundering transaction behaviors related to time and amount characteristics. CubeFlow money laundering detection method considers the density of node degrees along with amount characteristics but overlooks the temporal aspects of money laundering. While HoloScope comprehensively considers the characteristics of node degrees, temporal sequences, and the distribution of anomalous edges, it does not account for the transaction amount characteristics of anomalous edges. In summary, our DenseFlow method takes into account multiple features, including density, frequency, and transaction anomaly scores, enabling the detection of Ethereum money laundering with high precision and MCR.

## 6.3 RQ2: Suspiciousness Metric Assessment

To answer RQ2 and evaluate the three metrics we proposed, we conduct experiments individually with the three metrics (topological suspiciousness $\alpha$, temporal suspiciousness $\beta$, monetary suspiciousness $\gamma$) and their combinations.

The experiment results are shown in Figure 2. We get the following observations:

Observing precision results, we find that the $\gamma$ metric significantly improves the precision result in the CryptopiaHack case.

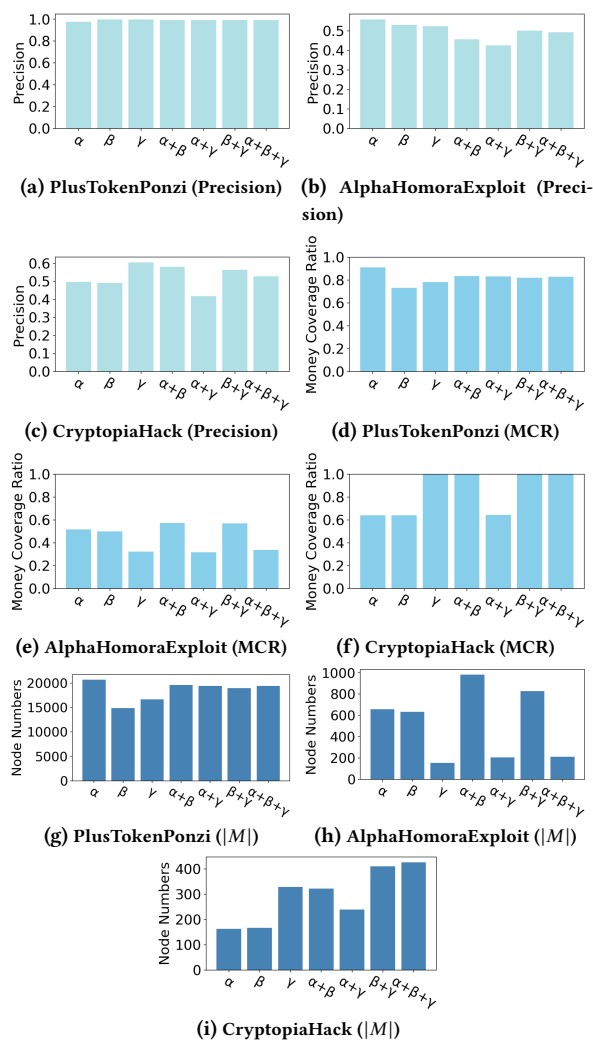

(a) PlusTokenPonzi (Precision)

(b) AlphaHomoraExploit (Precision)

(c) CryptopiaHack (Precision)

(d) PlusTokenPonzi (MCR)

(e) AlphaHomoraExploit (MCR)

(f) CryptopiaHack (MCR)

(g) PlusTokenPonzi ($|M|$)

(h) AlphaHomoraExploit ($|M|$)

(i) CryptopiaHack ($|M|$)

Figure 2: Results of DenseFlow and its variants.

This suggests that the density of laundering accounts in the CryptopiaHack case is significantly higher in the amount of money laundered and easy for us to catch. In the AlphaHomoraExploit case, the $\alpha$ metric has a more obvious enhancement effect on the precision results, which indicates that the interactions between the laundering accounts in this case are more intensive and more pronounced in Trait 1.

Observation of the MCR reveals that adding the $\beta$ metric in AlphaHomoraExploit improves the results, implying that the core money laundering accounts, in this case, have frequent transactions and significant temporal surges. In the PlusTokenPonzi case, the $\alpha$ metric is better portrayed, implying that the core accounts have more dense transfers.

Looking at the number of identified nodes $|M|$, we find that considering the $\gamma$ metric helps narrow the scope better with fewer nodes in the AlphaHomoraExploit case, where the core gang can be effectively differentiated with the $\gamma$ metric. In contrast, $\alpha$ and $\beta$ metrics play a similar role in the CryptopiaHack case.

In general, since laundering accounts in different cases have different preferences and may use different strategies to avoid detection, the actual algorithm can be based on the needs of a comprehensive consideration of the metrics to be used.

## 6.4 RQ3: Case Study

To answer RQ3, we visualize the traced money laundering pathways in DenseFlow. We take a money laundering pathway on the AlphaHomoraExploit dataset as an example, as shown in Figure 3.

The source node represents the origin hacker account of this incident. Nodes in $S^*$ are suspicious accounts identified through Algorithm 1. Nodes in $F$ represent accounts on the pathway between suspicious accounts and source accounts found by the maximum flow algorithm. The arrow color intensity indicates the proportion of money laundering amounts in the entire flow, with darker colors indicating higher proportions and lighter colors indicating lower proportions.

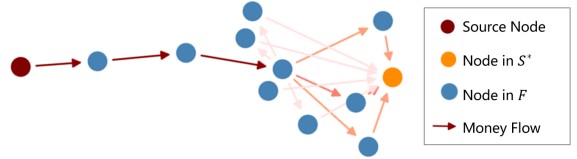

Figure 3: Visualization of an identified money laundering pathway on the AlphaHomoraExploit dataset.

We observe that the money laundering pathways evolve from sparse to dense. A clear flow pattern emerges in the upstream layers (around 1-3 layers), indicating significant transactions. However, as we move downstream, the number of transactions becomes denser, and the money laundering pathways become more complex, even forming loops. Money laundering amounts are continuously diluted, with more dispersion into smaller transactions as we approach downstream. The graph shows this in the lighter colors of transactions in the downstream part of the pathways. The money laundering pathways eventually converge from multiple dispersed accounts to the heist account.

## 7 CONCLUSIONS

This paper proposes an Ethereum money laundering detection method called DenseFlow. DenseFlow considers three suspiciousness metrics according to money laundering traits and supplements the laundering pathways via maximum flow. DenseFlow offers several advantages: 1) Suspiciousness metrics proposal and adaptable strategies: We introduce a novel transaction rating and provide different account metrics combinations. These strategies can be selected depending on the specific characteristics of different cases. 2) Effectiveness: DenseFlow outperforms existing algorithms regarding precision and money cover ratio. DenseFlow can identify core account nodes with higher precision and higher money coverage. We enable clear tracking of fund flows, aiding in case investigation and evidence gathering. 3) Interpretability. It searches for dense subgraphs based on well-defined descriptive features and supplements their connections using the maximum flow algorithm. The underlying model logic is straightforward and transparent.

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

# A APPENDIX

## A.1 Proof for LEMMA 2

LEMMA 2. *For optimal subset $S^*$ and any node $i \in S^*$:*

$$f_i(S^*) \geq g(S^*)$$

PROOF. Suppose there exists node $i_0$ such that

$$f_{i_0}(S^*) < g(S^*)$$

then we consider subset $S' = S^* \backslash \{i_0\}$, and

$$\begin{aligned}
g(S') = \frac{f(S')}{|S'|} &= \frac{f(S')}{|S^*| - 1} \\
&\geq \frac{f(S^*) - f_{i_0}(S^*)}{|S^*| - 1} \quad \text{By Lemma 1} \\
&\geq \frac{f(S^*) - g(S^*)}{|S^*| - 1} \quad \text{By the suppose} \\
&= \frac{f(S^*) - g(S^*)}{|S^*| - 1} = g(S^*)
\end{aligned}$$

There is a contradiction with the condition that $S^*$ is the optimal subset. Hence, the assumption is not valid and the original proposition is proven. $\square$

## A.2 Proof for Inequality (13)

Continuing with the labeling used in the main text, $S^* = S^{(k)}$, and denote $S^{(k-1)}$ as the last subset before $S^{(k)}$, and $i_1$ represents the node that is removed at the last step, that is $S^* = S^{(k-1)}/\{i_1\}$.

Proof.

$$g(S^{(k-1)}) \geq \frac{(b-1)}{b^2} f_{i_1}(S^{(k-1)})$$

Let the $b$-quantile of $\{f_i(S)|i \in S\}$ denoted as $Q_b(S)$, and the corresponding nodes are denoted as $i_b$. To simplify notation, denote $p = \frac{1}{b}$.

$$\begin{aligned}
g(S^{(k-1)}) &= \frac{1}{|S^{(k-1)}|} \sum f_i(S^{(k-1)}) \\
&= \frac{1}{|S^{(k-1)}|} \left( \sum_{f_i \geq Q_b(S^{(k-1)})} f_i(S^{(k-1)}) + \sum_{f_i < Q_b(S^{(k-1)})} f_i(S^{(k-1)}) \right) \\
&\geq (1-p)Q_b(S^{(k-1)}) + \frac{1}{|S^{(k-1)}|} \sum_{f_i < Q_b(S^{(k-1)})} f_i(S^{(k-1)}) \\
&\geq (1-p)Q_b(S^{(k-1)}) \\
&= (1-p)\frac{Q_b(S^{(k-1)})}{f_{i_1}(S^{(k-1)})} f_{i_1}(S^{(k-1)})
\end{aligned}$$

Now we consider $\frac{Q_b(S^{(k-1)})}{f_{i_1}(S^{(k-1)})}$, according to the definition of $f_i(S)$,

$$\begin{aligned}
\frac{Q_b(S)}{f_{i_1}(S)} &= \frac{\sum_{(j,i_b) \in E \wedge j \in S} e_{j i_b} \cdot b^{\alpha_{i_b}(S) + \beta_{i_b}(S) + \gamma_{i_b}(S) - 3}}{\sum_{(j,i_1) \in E \wedge j \in S} e_{j i_1} \cdot b^{\alpha_{i_1}(S) + \beta_{i_1}(S) + \gamma_{i_1}(S) - 3}} \\
&\geq \frac{1}{b} \frac{\sum_{(j,i_b) \in E \wedge j \in S} e_{j i_b}}{\sum_{(j,i_1) \in E \wedge j \in S} e_{j i_1}}
\end{aligned}$$

Since $i_1$ is the node with maximum weight.

$$\geq \frac{1}{b}$$

Therefore,

$$\frac{Q_b(S^{(k-1)})}{f_{i_1}(S^{(k-1)})} \geq \frac{1}{b}$$

Furthermore,

$$\begin{aligned}
g(S^{(k-1)}) &\geq (1-p) \cdot \frac{1}{b} f_{i_1}(S^{(k-1)}) \\
&= (1 - \frac{1}{b}) \cdot \frac{1}{b} f_{i_1}(S^{(k-1)}) \\
&= \frac{(b-1)}{b^2} f_{i_1}(S^{(k-1)})
\end{aligned}$$

Therefore, the inequality holds, and the proof is complete. $\square$

Received 12 October 2023

