# OpenReview forum: "DenseFlow: Spotting Cryptocurreny Money Laundering in Ethereum Transaction Graphs"
_ACM.org/TheWebConf/2024/Conference — TheWebConf24_

### Official Review · Reviewer_9Xbb · 2023-11-09

**Novelty:** 4
**Technical Quality:** 4

**Review:**

This paper presents a new framework designed to assist regulars in identifying and evidencing money laundering activities on Ethereum. It designs the suspiciousness metric for accounts and transactions based on the traits of money laundering behavior. Multiple experiments are conducted to evaluate the effectiveness of the proposed method, and the experimental results verify the effectiveness.

The authors summarize four traits in money laundering activities and propose three suspiciousness metrics. However, the generality of the traits is doubtable. Moreover, hackers can easily evade the detection through changing their behaviors accordingly.

The construction of graph $G$ is not clearly described. In the paper, the graph is constructed from the hacker's downstream transactions. How to construct such a graph if we don't know who is the hacker? If we construct the graph from transactions in the wild, can topoloigical suspiciousness and temporal suspiciousness be calculated correctly since they are closely related to he size of $V$?

The presentation of this paper can be improved. Many symbols miss explanation for important equations.

**Questions:**

Can the method detect money laundering if the hackers transfer the money in a very slow rate?

**Ethics Review Description:**

No ethical issues

**Reviewer Confidence:**

3: The reviewer is confident but not certain that the evaluation is correct

**Scope:**

4: The work is relevant to the Web and to the track, and is of broad interest to the community

---

### Official Review · Reviewer_bowM · 2023-11-20

**Novelty:** 6
**Technical Quality:** 6

**Review:**

This paper propose an innovative DenseFlow framework that effectively identifies and traces money laundering activities on blockchain by finding dense subgraphs and applying the maximum flow idea. Evaluation results show that the proposed framework can achieve significant improvement over existing methods.

## Pros
+ This paper tackles a very important problem in the Web community.

+ The challenges of the problem is well summarized. Moreover, the insights of the proposed approach to tackle these challenges are well described. The approach is sound to me.

+ The research questions and evaluation settings are well justified.

+ The improvement over the state-of-the-art is significant.

+ The paper is well written and easy to follow.

Overall, I like this paper and I am happy to see this paper accepted.

## Cons

I do not see obvious cons of this paper

**Questions:**

I do not have specific questions at this stage.

**Reviewer Confidence:**

2: The reviewer is willing to defend the evaluation, but it is likely that the reviewer did not understand parts of the paper

**Scope:**

4: The work is relevant to the Web and to the track, and is of broad interest to the community

---

### Official Review · Reviewer_3p6G · 2023-11-21

**Novelty:** 4
**Technical Quality:** 5

**Review:**

This paper explores the problem of detecting money laundering accounts in Ethereum transaction graphs.  It was nicely written, easy to follow, has clear motivation, and the results seemed strong.  I have a few concerns, below:

1. I would have liked to see some more discussion of how exactly results could be used.  The goal of identifying such fraud clearly seems like an important goal.  But given that accounts are anonymous and easy to create, how does it help?  Can they be frozen in real time?  An end-to-end use case example would be helpful, even if given at a high-level.

2. The problem statement is quite informal.  For example, does "consisting of suspicious..." mean "consisting only of suspicious..."?  "Consisting mostly of suspicious..."?  Something else?

3. Why is it reasonable to think that the groups of suspicious accounts are dense, given that it's so easy to make new accounts?  Why wouldn't there just be a new node for each transaction?  But looking at Eq. 1, it doesn't appear that you're actually looking at the density of the subgraph S-- you're looking at what fraction of each node's connections lie within that subgraph.  This is a bit different than the traditional definition of density.

4. I didn't understand the max flow part of the algorithm.  Why are you finding max flows using the transaction amounts as the capacities?  Don't the transactions represent actual flow, as opposed to an upper bound on allowable flow?

5. The Lin [22] paper seems to play an important role here, including in the experimental analysis.  It needed more discussion, including discussion of how accurate it is.

Thanks to the authors for their responses and clarifications.

**Questions:**

Please respond to my concerns in #4 and #5, specifically.

**Ethics Review Description:**

-

**Reviewer Confidence:**

3: The reviewer is confident but not certain that the evaluation is correct

**Scope:**

3: The work is somewhat relevant to the Web and to the track, and is of narrow interest to a sub-community

---

### Official Review · Reviewer_DVSE · 2023-11-24

**Novelty:** 4
**Technical Quality:** 4

**Review:**

This article introduces an innovative DenseFlow framework that employs a greedy algorithm to identify dense subgraphs and applies the concept of maximum flow to effectively detect and track money laundering activities.

Advantages:
- This paper introduces several interesting ideas to quantify the suspiciousness of accounts.
- Its logic is clear and makes for a pleasant read.

Disadvantages:
- The objective function is somewhat subjective, and its specific design awaits experimental validation.
- The employed methods (greedy algorithm and maximum flow) lack innovativeness.

**Questions:**

With the development of deep learning, can deep learning methods be employed to optimize the objective function? In this context, given an optimization function, the goal is to find a subgraph from the graph that maximizes the optimization function, and utilizing graph networks could be considered as a potential implementation.

**Reviewer Confidence:**

1: The reviewer's evaluation is an educated guess

**Scope:**

2: The connection to the Web is incidental, e.g., use of Web data or API

---

### Decision · Program_Chairs · 2024-01-22

**Decision:**

Accept

**Comment:**

Thank you for submitting your paper to The Web Conference 2024. Your rebuttal has addressed most of the concerns raised by the reviewers. At this stage, I do not see clear reasons that block the acceptance of the paper.

 We kindly request that you carefully address the concerns raised by the reviewers in the final version of your paper.